# Quality of life and metabolic outcomes after total pancreatectomy and simultaneous islet autotransplantation

Stefan Ludwig[1,8], Marius Distler[1,2,3,8], Undine Schubert[2,3,4], Janine Schmid[2,3,4], Henriette Thies[4], Thilo Welsch[1], Sebastian Hempel[1], Torsten Tonn [5,6], Jürgen Weitz[1,2,3], Stefan R. Bornstein [2,3,4,6,7] & Barbara Ludwig [2,3,4,6✉]

## Abstract

**Background** Pancreas surgery remains technically challenging and is associated with considerable morbidity and mortality. Identification of predictive risk factors for complications have led to a stratified surgical approach and postoperative management. The option of simultaneous islet autotransplantation (sIAT) allows for significant attenuation of long-term metabolic and overall complications and improvement of quality of life (QoL). The potential of sIAT to stratify a priori the indication for total pancreatectomy is yet not adequately evaluated.

**Methods** The aim of this analysis was to evaluate the potential of sIAT in patients undergoing total pancreatectomy to improve QoL, functional and overall outcome and therefore modify the surgical strategy towards earlier and extended indications. A center cohort of 24 patients undergoing pancreatectomy were simultaneously treated with IAT. Patients were retrospectively analyzed regarding in-hospital and overall mortality, postoperative complications, ICU stay, hospital stay, metabolic outcome, and QoL.

**Results** Here we present that all patients undergoing primary total pancreatectomy or surviving complicated two-stage pancreas resection and receiving sIAT show excellent metabolic outcome (33% insulin independence, 66% partial graft function; HbA1c 6,1 ± 1,0%) and significant benefit regarding QoL. Primary total pancreatectomy leads to significantly improved overall outcome and a significant reduction in ICU- and hospital stay compared to a two-stage completion pancreatectomy approach.

**Conclusions** The findings emphasize the importance of risk-stratified pancreas surgery. Feasibility of sIAT should govern the indication for primary total pancreatectomy particularly in high-risk patients. In rescue completion pancreatectomy sIAT should be performed whenever possible due to tremendous metabolic benefit and associated QoL.

## Plain language summary

Pancreas surgery is complicated and associated with substantial risks and even danger of death. The surgical removal of the whole pancreas can be necessary for some indications but results in a severe form of diabetes. The method of islet autotransplantation (IAT) involves taking the pancreas, isolating the insulin-producing cells and returning these to the patient. This helps to preserve insulin production and minimises the impact of diabetes. We retrospectively analyzed a cohort of patients undergoing pancreatectomy that were simultaneously treated with IAT. The analysis included short-term and long-term surgical and diabetes-related outcomes as well as quality of life. All parameters indicated the benefit of IAT in patients that require extensive pancreas surgery. Offering IAT to patients may reduce surgical complications after pancreatectomy, enhance recovery, and therefore facilitate faster initiation of other therapies where needed.

[1] Department of Visceral, Thoracic and Vascular Surgery, University Hospital Carl Gustav Carus Technische Universität Dresden, D-01307 Dresden, Germany. [2] Paul Langerhans Institute Dresden of the Helmholtz Center Munich at The University Hospital Carl Gustav Carus and Faculty of Medicine of the Technische Universität Dresden, D-01307 Dresden, Germany. [3] Center for Diabetes Research (DZD e.V.), D-85764 Neuherberg, Germany. [4] Department of Medicine III, University Hospital Carl Gustav Carus, Technische Universität Dresden, D-01307 Dresden, Germany. [5] Experimental Transfusion Medicine, Faculty of Medicine of the Technische Universität Dresden, D-01307 Dresden, Germany. [6] DFG-Center for Regenerative Therapies Dresden, Technische Universität Dresden, D-01307 Dresden, Germany. [7] Division of Diabetes & Nutritional Sciences, Faculty of Life Sciences & Medicine, King's College London, London SE1 1UL, UK. [8] These authors contributed equally: Stefan Ludwig, Marius Distler. ✉email: Barbara.ludwig@uniklinikum-dresden.de

Traditionally, islet autotransplantation (IAT) has been performed in patients undergoing pancreatectomy for chronic pancreatitis[1–3]. Increasing evidence emerges that IAT simultaneously performed with total or subtotal pancreatic surgery is a beneficial strategy for extended indications beyond chronic pancreatitis that may lead to reduced complications, improved metabolic outcome, and quality of life (QoL)[4–6]. Indeed, the option of islet isolation and autologous re-transplantation may shift the indication towards primary total pancreatectomy and avoid relaparatomies/rescue pancreatectomies and complications in borderline or high-risk patients.

Pancreas surgery remains technically challenging and is associated with considerable morbidity and mortality for all indications[7,8]. Particularly, postoperative pancreatic fistula (POPF) is the single most common complication after major partial pancreatic resections[9–11] and the concomitant complications (e.g. postoperative pancreatic hemorrhage, PPH) significantly increases the perioperative mortality[12]. Even if acute complications were prevented or overcome, the loss of pancreatic endocrine and exocrine parenchyma leads to the development of pancreatogenic diabetes. This diabetes type ranks among the worst forms of diabetes due to the concurrent glucagon deficiency and is further complicated by exocrine deficiency. Metabolic lability and severe hypoglycemia are typical complications in this patient group. Moreover, suboptimal metabolic control causes long-term complications and there is increasing evidence, that metabolic control is significantly associated with the oncological outcome[13]. IAT is a viable option to prevent pancreatogenic diabetes or reduce the severity of diabetes after total pancreatectomy[4]. Pancreatic islets are isolated from resected pancreas tissue by enzymatic digestion, optionally purified from exocrine tissue and intraportally grafted. In this analysis, we report on our experience with IAT established in a high-volume pancreas center in patients undergoing pancreas surgery for various indications from non-malignant, borderline, and malignant diseases to salvage pancreatectomy due to complications following pancreaticoduodenectomy and pancreas involving abdominal trauma. Our data demonstrate the feasibility, efficacy, and safety of IAT in this patient cohort and may stimulate further evaluation of this strategy to improve overall outcome, metabolic control, and QoL after pancreatic surgery.

## Methods

**Patients**. From 05/2012 to 08/2018 patients scheduled for pancreatic surgery at the Department of Visceral,- Thorax- and Vascular Surgery at the TU Dresden were evaluated for eligibility for IAT (Table 1). The following conditions ($n = 29$) were included: (1) chronic pancreatitis and exhausted conservative treatment, (2) traumatic pancreas rupture, (3) severe complications after pancreatic surgery requiring completion pancreatectomy, (4) primary or completion pancreatectomy following duodenal ulcers/adenoma, (5) duodenal or papillary tumor, (6) localized IPMN, and (7) highly differentiated neuroendocrine tumor and sarcoma in the pancreas head (Table 2). Prior to surgery, all patients were assessed by contrast-enhanced computer tomography scan and transabdominal/endoscopic ultrasound, partially with fine needle aspiration. According to individual indications, additional studies including Magnetic Resonance Imaging and Positron Emission Tomography were performed.

**Indication for islet transplantation**. According to the center licence for pancreas retrieval and utilization for islet processing and transplantation, the indication for IAT was chronic pancreatitis when subtotal or total pancreatectomy was indicated (failed medical therapy), patients undergoing pancreaticoduodenectomy where pancreatic anastomosis was assessed as high risk for leakage (combination of narrow duct and soft and/or frail pancreatic parenchyma), severe complications after pancreatic surgery; grade C pancreatic fistula (according to the definition of the International Study Group on Pancreatic Fistula) requiring relaparotomy with completion pancreatectomy. General exclusion criteria were portal vein thrombosis, liver disease with reduction of liver function, multifocal pancreatic neoplasm at preoperative imaging or intraoperative evaluation, malignant disease where the pancreatic transection margin is involved (no confirmed margin negativity by immediate intraoperative pathology) including any degree of dysplasia or ductal dysepithelialization, diagnosis of multiple endocrine neoplasm, diabetes mellitus of any type with insulin deficiency that requires insulin treatment.

All patients were treated following center standards according to the respective underlying disease.

**Islet processing and transplantation**. Islets were isolated and optionally purified from resected pancreas tissue according to a modified Ricordi method[14]. Briefly, Collagenase, neutral protease (Serva Electrophoresis, Heidelberg, Germany), and Pulmozyme (Roche, Grenzach, Germany,) were infused into the main pancreatic duct. Islets were optionally separated from exocrine tissue by centrifugation on a continuous Biocoll gradient (Biochrom AG, Berlin, Germany) in a COBE 2991 cell processor (Lakewood, CO, USA). Isolation characteristics are summarized in Table 2. Negative gram stain and endotoxin measurement were defined as mandatory release criteria immediately before transplantation. Additionally, bacterial culture was performed for 14 days in every preparation. Strict release criteria as established for allogeneic islet transplantation regarding islet viability, yield, and purity were not generally defined. However, an islet yield <100 IEQ/kg BW was considered as failed isolation procedure and as not transplantable.

Islets were transplanted at the same day according to the surgical, clinical, and logistic situation either intraoperatively or after a short time interval at the ICU unit. Islet transplantation was performed by direct puncture of the portal vein, introduction of a triple-lumen catheter, and under continuous monitoring of portal vein pressure. Heparin (5000 IU) was added to the islet preparation, immediately postoperatively patients were started on a continuous intravenous heparin drip, monitored and adjusted according to activated partial thromboplastin time (PTT; range of 40–50 s) for at least 48 h and thereafter switched to subcutaneous low molecular weight heparin until discharge. Target PTT was discussed with the surgeons and adjusted where appropriate[15].

**Perioperative management and follow-up**. Intraoperatively patients were started with i.v. insulin infusion with target blood glucose levels between 4 and 8 mmol/l and continued during at least 3 postoperative days. According to clinical conditions (no

### Table 1 Patient characteristics.

| | |
|---|---|
| Number of cases evaluated | $n = 52$ |
| Number of realized islet isolations | $n = 35$ |
| Number of transplantations | $n = 24$ |
| Median follow-up (months) | 46 ± 5.7 |
| Age (years) | 57 ± 23 |
| Sex (m/f) | 14/10 |
| BMI (kg/m²) | 24.9 ± 4.7 |
| Weight (kg) | 72.3 ± 21.6 |
| Fasting plasma glucose (mmol/l) | 4.6 ± 0.8 |
| Impaired fasting glucose | 2/24 (8%) |
| HbA1c (%) | 5.2 ± 0.3 |

**Table 2 Diagnosis and transplant characteristics.**

|  | sIAT after primary total pancreatectomy | sIAT after completion pancreatectomy |
|---|---|---|
| Patients (n) | 14 | 10 |
| Diagnosis |  |  |
| Chronic pancreatitis | 4 | 3 |
| Abdominal trauma | 1 | 1 |
| Pancreatic cystic neoplasm | 2 | – |
| NET (Grade 2) | 1 | – |
| Dendritic cell sarkoma | 1 | – |
| Mesenchymal mediastinal sarkoma | 1 | – |
| Non pancreas related |  |  |
| Duodenal adenoma/ulcer | 2 | 3 |
| Mesenteric ischemia | 1 | – |
| Duodenal carcinoma | 1 | 1 |
| Papillary tumor | – | 2 |
| Trimmed pancreas weight (g) | 85 ± 39[*] | 51 ± 24[*] |
| Purification (y/n) | 14/0 | 7/3 |
| Islet yield [IEQ (x10³)] | 256 ± 93 | 214 ± 190 |
| Islet yield (IEQ/g) | 3590 ± 2680 | 4798 ± 5252 |
| Purity (%) | 67 ± 15 | 57 ± 26 |
| Pre-transplant culture (y/n) | 0/14 | 1/9 |
| Endotoxin/microbiology positive (y/n) | 0/14 | 0/10 |
| Islets infused (IEQ/kg BW) | 3351 ± 676 | 2618 ± 2516 |

Significant difference between the groups.
[*]$p < 0.05$.

pressors, oral feeding), therapy was switched to a subcutaneous insulin treatment regimen adapted to metabolic requirements. Islet function was assessed by measurement of fasting C-peptide, blood glucose measurements, and daily insulin requirement. In case of malignant diseases, adjuvant chemotherapy or radio-therapy was administered when indicated, imaging and serum marker testing were performed as per center standard.

**Hospital and ICU stay.** For all patients included in this analysis, total hospital stay and duration of ICU treatment were comparatively calculated for sIAT after primary total pancreatectomy after completion pancreatectomy.

**Diabetes therapy.** Patients were evaluated at least every three months regarding glycemic control and insulin therapy was adjusted or discontinued. Insulin independence was defined as adequate glycemic control (HbA1c <7%, fasting blood glucose levels <6.9mmol/l, and 2-h postprandial blood glucose levels <10 mmol/l) without insulin therapy[4,16,17]. Partial graft function was defined as fasting c-peptide level >0.26 nmol/l and need for supportive exogenous insulin to achieve adequate glycemic control. Primary graft non-function or graft loss was defined as fasting C-peptide levels <0.26 nmol/l after islet infusion or during follow-up respectively.

**Quality of life assessment.** All patients were assessed using the validated Diabetes Distress (DD) Score[18] that includes a total number of 28 questions regarding seven sources of DD among adults that are critically related to a variety of patient demographic and disease-related characteristics: Feel of powerlessness, difficulties in handling the diabetes, problems with hypoglycemia, social burden, eating-related burden, physician-related burden, family/friends related burden. The assessment was performed according to Fisher et al[18]. by calculation of a mean item-score that is then grouped into three categories: 1,0: no or minimal diabetes-associated distress; 1,0–1,9: moderate diabetes-associated distress; ≥2,0: increased diabetes-associated distress. As control group, an equal number of patients that underwent standard

surgical treatment without islet auto-transplantation resulting in total pancreatectomy and insulin deficiency were identified and matched with regard to age, gender, malignant/non-malignant underlying disease.

**Ethics approval.** The retrospective study was approved by the TU Dresden Institutional Review Board (EK 310062019) with written informed consent obtained from each participant. Additionally, all patients included in this study have provided written informed consent as required within our center licence for islet transplantation and consented to scientific data analysis. Approval for QoL assessment was obtained from the Ethics committee of the University of Dresden, Germany on June 12 2017 (EK 255062017), University of Zurich, Switzerland on April 20th 2015 (KEK-ZHNo: 2014–0631).

**Statistical analysis.** Data processing was performed using GraphPad Prism 4 (Graph-Pad Software, La Jolla, CA, USA). Student's $t$ test was used to establish comparisons between groups. Results are shown as mean ± SD. Survival was estimated according to Kaplan-Meier analysis. Significance levels were established at a $p$-value of <0.05.

**Reporting summary.** Further information on research design is available in the Nature Research Reporting Summary linked to this article.

## Results

**Patients, surgery, overall follow-up.** A total number of 52 patients were evaluated for the possibility of IAT simultaneous with pancreatectomy. In 35 cases islet isolation from resected pancreas tissue was performed and in 24 patients transplantation could be realized. The main reason for cancellation of islet isolation was pre-existing insulin-dependent diabetes, the presence of multifocal pancreatic neoplasm in preoperative imaging or intraoperative evaluation, or insufficient or highly fibrotic/necrotic pancreas tissue. In 11 cases islet isolation was not successful (insufficient number of islets; IEQ/kg < 100) and

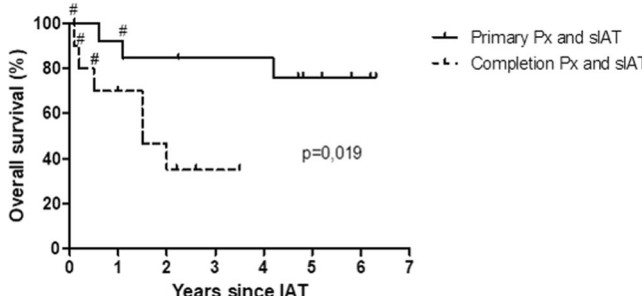

**Fig. 1 Follow-up: overall survival.** Probability of overall survival after pancreatectomy combined with sIAT either as primary total ($n = 14$) or completion pancreatectomy ($n = 10$) according to Kaplan–Meier. During median follow up of $46 \pm 5.7$ months, three patients died in the primary pancreatectomy group and seven deaths occurred in the completion pancreatectomy group. # represents in-hospital death. Px pancreatectomy, sIAT simultaneous islet autotransplantation.

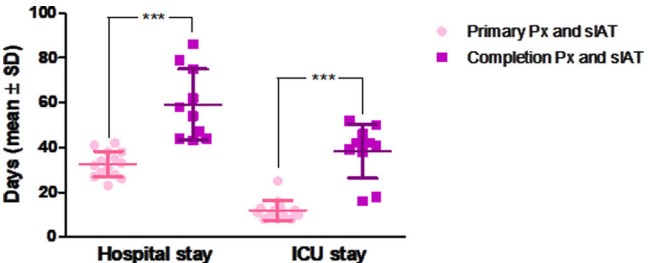

**Fig. 2 Postoperative course following primary total pancreatectomy ($n = 14$) and completion pancreatectomy ($n = 10$).** The hospital stay was significantly longer in patients with completion pancreatectomy compared to primarily pancreatectomized patients ($59 \pm 15$ versus $32 \pm 5$ days). ICU-time was equally significantly prolonged in the completion pancreatectomy group ($38 \pm 12$ versus $12 \pm 4$ days). Data are shown as individual data points and mean $\pm$ SD; Variables were compared with the Student's $t$ test; ***$p < 0.001$.

transplantation must be called off. From the 24 realized cases, 14 patients were treated by primary total pancreatectomy, 10 patients underwent salvage completion pancreatectomy due to complications (grade B/C POPF) following pancreaticoduodenectomy. Patient characteristics are shown in Table 1.

In the 24 cases, where islet transplantation was realized, no complications related to IAT were observed. Except for one case, islet infusion did not relevantly affect the portal vein pressure (increase < 50% of baseline). In this one case, we saw a steady increase up to 50% of baseline after two third of islet infusion. The procedure was therefore stopped and the remaining islet mass was grafted into an ad hoc prepared omental pouch. The median follow-up was $46 \pm 5.7$ months, and 14 (58.3%) of 24 patients were still alive (Fig. 1). Three deaths occurred in the primary pancreatectomy group: Two patients died from septic complications and multiorgan failure (in-hospital death) with duodenal ulcer and duodenal cancer respectively as underlying diseases. One death was unrelated to the primary disease (chronic pancreatitis). This patient died from pulmonary embolism 30 months after initial surgery. Seven patients died in the completion pancreatectomy group: two patients developed fatal septic complications (in-hospital death), three patients died from disease progression, one patient died from gastrointestinal bleeding (related to multiple interventions due to mesenteric ischemia), and one patients died after rupture of a thoracic aortic aneurysm unrelated to the initial disease.

As shown in Fig. 2, patients that underwent primary total pancreatectomy had a significant shorter ICU-time compared to patients that were treated by relaparatomy and salvage pancreatectomy ($12 \pm 4$ days vs. $38 \pm 12$ days, respectively). Also the total hospital stay was significantly reduced in the primary pancreatectomy group compared to the salvage pancreatectomy group ($32 \pm 5$ days vs. $59 \pm 15$ days, respectively).

**Islet isolation outcome and metabolic follow-up.** The results for islet isolation and transplantation are summarized in Table 2 stratified for primary or salvage pancreatectomy. As to be expected, the pancreas tissue available for islet isolation was considerably larger when patients were primarily totally pancreatectomized ($85 \pm 39$ g vs. $51 \pm 24$ g in completion pancreatectomy group). However, islet yield ($256,000 \pm 93,000$ IEQ vs. $214,000 \pm 190,000$ IEQ), purity ($67 \pm 15\%$ vs. $57 \pm 26\%$), and infused islet mass ($3351 \pm 676$ IEQ/kgBW vs. $2618 \pm 2516$ IEQ/kgBW) was not different between the groups. The islet yield per gram of pancreas tissue was in fact relatively higher in the salvage pancreatectomy group ($3590 \pm 2680$ IEQ/g vs. $4798 \pm 5252$ IEQ/g). Unpurified islets were only infused in three cases, all of them

in patients with non-malignant diseases and low islet yield. In all islet preparations sterility testing was negative as tested by pre-transplant gram stain and endotoxin assay as well as microbial culture (results available only post transplantation).

During follow-up, eight (33%) of 24 patients reached insulin independence; the other 16 (66%) patients had partial graft function (Fig. 3). At the last follow-up visit, fasting C-peptide was $0.53 \pm 0.29$ nmol/l, HbA1c was $6.1 \pm 1.0\%$, and insulin requirement was $0.09 \pm 0.1$ IU/kg/d. In patients with exogenous insulin need, the treatment regimen varied between solely basal insulin, on-demand prandial insulin, and MDI-therapy. The treatment goals were adapted to individual patient situation with primary goal of avoidance of hypoglycemic episodes. The latter was reached in all patients followed.

**Assessment of quality of life.** QoL after total pancreatectomy and sIAT was analyzed using a validated questionnaire (Diabetes Distress Score; DDS) that tested seven sub-categories of diabetes-related distress by a total of 28 questions. For comparison, matched patients undergoing standards treatment without IAT were questioned. As shown in Table 3, patients with sIAT showed a mean item-score between 1,0 and 1,9 (moderate diabetes-associated distress) in all categories except for the category 'feel of powerlessness', where patients ranked 2.18 (increased diabetes-associated distress). This overall positive result was seen regardless of primary or completion pancreatectomy and was dependent on the presence of IAT. In contrast, the matched control group with complete insulin deficiency and solely insulin therapy resulted in item-scores between 1,0 and 1,9 (moderate diabetes-associated distress) only in the categories 'social burden' and 'physician related burden'. All other categories were categorized as increased diabetes-associated distress ('Difficulties in handling the diabetes', 'Eating related burden', 'Family/friends related burden') or even highly increased distress ('Feel of powerlessness', 'Problems with hypoglycemia').

## Discussion

The overall outcome after pancreas surgery including pancreas head resection (PPPD/Whipple) for oncological cases has improved considerably over the last decade. The improvement is mainly the result of better chemotherapy regimens and advancements in the surgical technique. Therefore, postoperative QoL and metabolic outcome become increasingly relevant[19–21]. Despite that pancreatic surgery is still associated with high morbidity (30–50%) and mortality rates (5–10%) mainly caused by postoperative pancreatic fistula (POPF) and the related complications (e.g. postoperative pancreatic hemorrhage (PPH))[12].

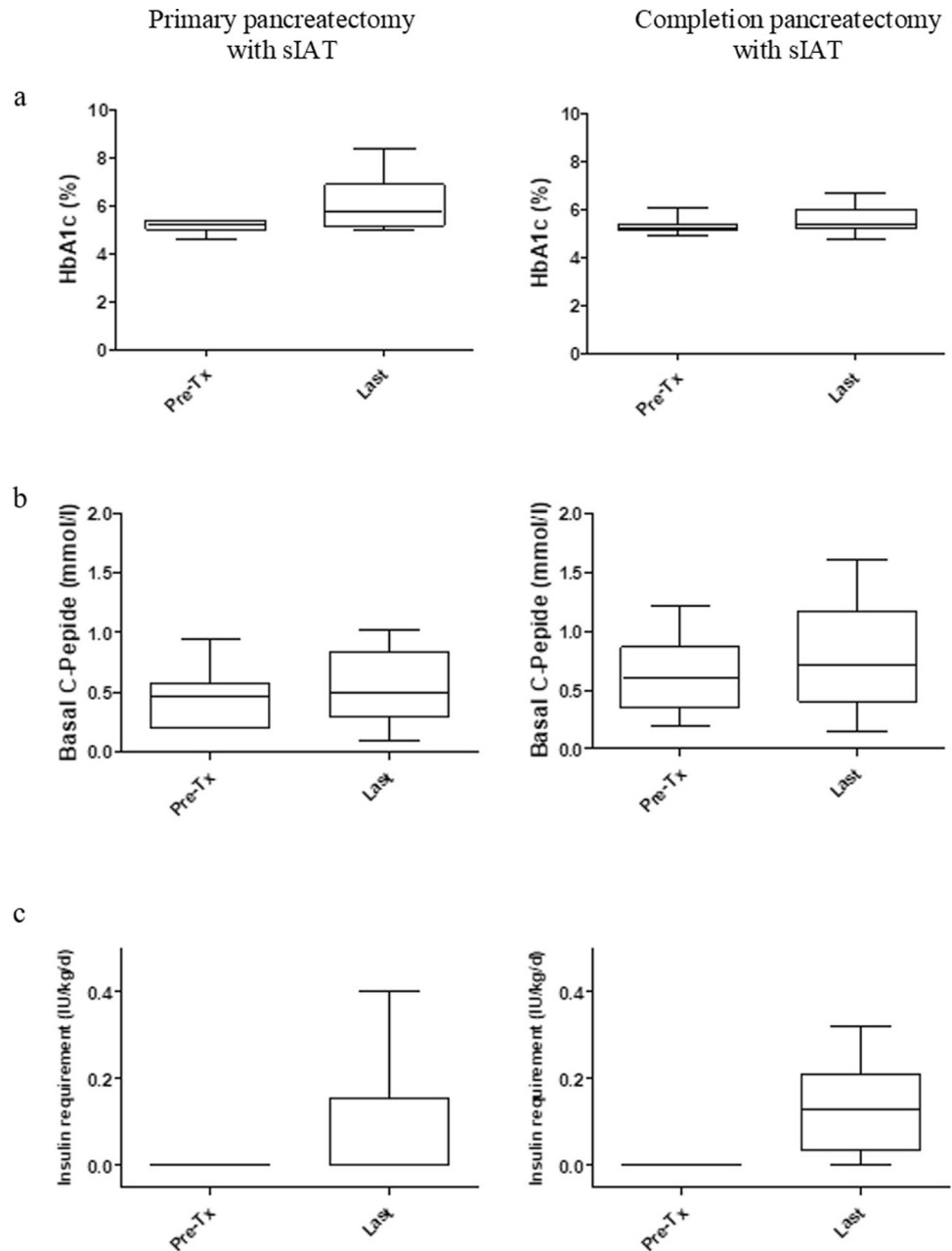

**Fig. 3 Metabolic control and graft function after IAT.** Box plots of glycated hemoglobin (HbA1c) (**a**), basal C-peptide (**b**), and insulin requirement (**c**) before and at the last follow-up visit after IAT. Data were analyzed separately for patients treated with primary total ($n = 14$) or completion pancreatectomy ($n = 10$) and are shown as mean ± SD.

Morbidity on the one hand is considerably increasing hospitalization and medical costs and leads to a deterioration in QoL. On the other hand and more importantly, high morbidity leads to a delay or even failure of adjuvant treatment in cancer patients and thereby has a strong impact on the oncological outcome.

The aim of this analysis was to investigate, whether the option of IAT may allow for extending the indication for total pancreatectomy by retrospective analysis of a center cohort of patients undergoing pancreas surgery. Particularly, we hypothesized that in patients with high risk profile for developing complications after partial pancreatectomy, the a priori total pancreatectomy with simultaneous IAT may reduce morbidity and at the same time balance the otherwise short- and long-term negative effects of total pancreatectomy regarding metabolic outcome and QoL.

Balzano et al. have shown in several pioneering studies that extending the indication for IAT beyond the traditional application in chronic pancreatitis is feasible and safe. However, despite the convincing experimental[22,23] and clinical data shown by the Milan group[4,24], the discussion remains, whether it is justifiable to perform islet isolation and re-transplantation in patients with underlying malignant diseases and whether the risk for inducing liver metastasis through intraportal islet transplantation is a relevant concern. Indeed, the isolation process enriches endocrine cells, but cannot guarantee total elimination of ductal cells. The concern of inadvertent infusion of malignant cells might be valid. Alternative transplantation sites such as the intramuscular or subcutaneous region might offer additional safety and have been proven feasible sites experimentally and clinically[25,26]. However, the standard procedure in patients with pancreatic masses is a pancreatic head resection (PPPD/Whipple)

**Table 3 Quality of life assessed by Diabetes Distress Score (DDS).**

|  | Number of questions (total of 28) | Patients after total pancreatectomy and IAT (n = 24) | Matched patients with total pancreatectomy (without IAT) (n = 24) |
|---|---|---|---|
| Feel of powerlessness | 5 | **2,18** | **3,49** |
| Difficulties in handling the diabetes | 4 | 1,42 | **2,08** |
| Problems with hypoglycemia | 4 | 1,79 | **3,28** |
| Social burden | 4 | 1,17 | 1,75 |
| Eating related burden | 3 | 1,61 | **2,88** |
| Physician related burden | 4 | 1,71 | 1,88 |
| Family/friends related burden | 4 | 1,42 | **2,7** |

Bold numbers indicate "increased diabetes-associated distress".
A total of 28 questions covering 7 sub-categories were tested and analyzed using a validated score system (Fisher et al.). 1,0: no or minimal diabetes-associated distress; 1,0–1,9: moderate diabetes-associated distress; ≥2,0: increased diabetes-associated distress. Patients receiving sIAT were compared to matched patients undergoing standard surgical treatment without IAT.

allowing pancreas parenchyma in situ. From an oncological point of view, this is equivalent or even more risky compared to replacement of an endocrine enriched islet preparation from the same tissue. Moreover, if a primary total pancreatectomy in certain high risk patients leads to shorter period until adjuvant therapy can be initiated or at all realized, this strategy may have a strong positive impact on the oncological outcome. With regard to QoL, IAT has been shown to be tremendously beneficial to patients[27–29]. Especially in a patient group that is affected by a malignant disease with all physical and psychological stress, the additional burden through a typically highly instable glucose metabolism and the obligatory intense insulin treatment regimen is highly relevant. Independent of complete insulin independence, glycemic stability was reliably achieved in all patients receiving IAT. Besides beta cell secretory capacity, the effect on hypoglycemia counterregulation and subsequent glucose stability is an essential measure for metabolic outcome. In this cohort, none of IAT recipients experienced severe hypoglycemic episodes during follow-up. Reliable protection from hypoglycemia is not always achieved in the IAT setting (in contrast to allotransplantation in Diabetes mellitus type 1). This might be explained by the exclusively intrahepatic localized alpha-cells: During hypoglycemia, endogenous glucose production is induced and consequently leads to an increase in local hepatic glucose levels that are higher than the peripheral levels which in turn may inhibit adequate activation of intrahepatic alpha-cells[30,31]. Another hypothesis for the occurrence of post-prandial hypoglycemia after IAT is alimentary hypoglycemia due to the gastrointestinal reconstruction following pancreatectomy ("dumping syndrome")[32]. The absence of hypoglycemia in our patient cohort might be in part explained by routinely provided nutrition counseling that focuses on adapted dietary composition and frequent small meals.

We acknowledge, that this center cohort analysis overlooks a limited number of heterogeneous patients and ideally, randomized trials with stratified patient groups are needed to draw valid conclusions. However, this study brings several important insights in the treatment of patients that require pancreas surgery. First, in patients with pancreas injury due to abdominal trauma or as a consequence of abdominal surgery primarily unrelated to the pancreas, it is mandatory to offer the option of IAT and therefore preserve endocrine function. Second, the vast majority of patients with chronic pancreatitis is sufficiently treated by gradual conservative regimens[33] and only rare and thoroughly and interdisciplinary evaluated patients are candidates for pancreas resection. In these cases, the feasibility of IAT is dependent upon the extent of fibrosis and preserved endocrine tissue. This can possibly only be judged intraoperatively. Third, patients with high-risk profile for the development of POPF generally seem to benefit from a primary pancreatectomy with simultaneous IAT regardless of an underlying benign, borderline, or malignant disease. Particularly in cancer patients avoiding complications and relaparotomy, faster recovery, and a shorter interval until initiation of adjuvant therapy has a significant impact on oncological long-term outcome and QoL[34–36]. It seems reasonable to particularly investigate this constellation in a controlled manner. We therefore plan to initiate a clinical trial that hypothesizes that avoiding an entero-pancreatic anastomosis in a high risk situation for POPF by performing a primary total pancreatectomy with simultaneous islet autotransplantation (IAtx) will shorten the time until adjuvant therapies and therefore improve patients´ short- and long-term outcome. This may lead to a general paradigm shift in pancreatic surgery. It must be respected that islet transplantation is a complex procedure that is only available in specialized centers in Europe and international. However, there are several successful examples (e.g. UK, Nordic Network, GRACIL, Milan-Italy) where remote centers for pancreas surgery are associated with a central islet isolation facility and therefore this procedure can be provided at multiple sites without the necessity of an own islet isolation laboratory. Increasing number of IAT and multicenter experience will also help to develop clearer definitions of patient and pancreas tissue characteristics predictive for successful isolation and transplantation outcome.

Moreover, there is increasing evidence that metabolic control in patients with malignant diseases plays a determining role in disease progression[13]. Research in this direction is becoming more and more meaningful and supports the strategy of optimally controlling glycemia in cancer patients after pancreas surgery. IAT can contribute considerably to this goal regardless of full or partial graft function.

Taken together our findings and the little but highly substantial investigations in the literature particularly from the Milan group, it seems appropriate to ask whether the option of autologous islet isolation and transplantation may provoke a shift in paradigm in pancreatic surgery. The risks associated with this procedure are minimal. In fact, in our cohort we have seen no single complication related to IAT. Therefore and due to the major impact on QoL and long-term metabolic outcome it should be obligatory to evaluate the option of IAT in patients indicated for pancreas resecting surgery. We strongly push the further evaluation of this strategy to widen the indication for primary complete pancreatectomy and simultaneous IAT particularly in high-risk patients with malignant disease and need for early adjuvant therapy to be studied in controlled clinical trials.

## Data availability

Source data for the figures are available as Supplementary Data 1. Additional data sets that support the findings of this study are available from the corresponding author upon reasonable request.

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

## Author contributions

All authors have made substantial contribution to this work. S.L., M.D., T.W., S.R.B., J.W., and B.L. have formed the concept and design of this work. S.L., M.D., T.W., J.W., S.H., and B.L. contributed to patient acquisition and performed surgery, pre-and postoperative patient management. U.S., J.S., T.T., and B.L. have performed islet isolation and quality control. S.L., M.D., S.R.B., J.W., H.T., S.R.B., J.W., and B.L. have analysed and interpreted the data. S.L., M.D., H.T., T.T., S.R.B., J.W., and B.L. have drafted and substantively revised the work. All authors have approved the submitted version and agreed both to be personally accountable for the author's own contributions and to ensure that questions related to the accuracy or integrity of any part of the work, even ones in which the author was not personally involved, are appropriately investigated, resolved, and the resolution documented in the literature.

## Funding

## Competing interests

The authors declare no competing interests.
