## [Peer Review File · Communications Medicine]

Reviewers' comments:

Reviewer #1 (Remarks to the Author):

The authors report their experience in the emerging field of autologous islet transplantation after pancreatectomy.

This retrospective study compares 14 patients treated with this complex one-stage procedure with 10 patients who received this treatment in two stages, i.e. after completion of a pancreatectomy for complications of the first procedure.

Remarks :

- please specify in the method if the matching procedure was random or selected with or without a propensity score
 - line 68 with the Balzano et al reference you could add the Sutherland et al article "Total Pancreatectomy and Islet Autotransplantation for Chronic Pancreatitis" (JACS 2011)
 - We have seen that in the cohort out of 35 isolated pancreas only 24 could be transplanted. It would be very useful to specify the criteria used for the release of the islet batches, in particular in terms of purity / viability and number of islets. We understand in the "results" section that an insufficient number of 100 IEQ / kg was used as a limit but please specify these criteria in the methods
 - please specify in the methods the anticoagulation protocol used for the intraportal autograft. Indeed, we could read in a very remaining article that the modalities of anticoagulation were heterogeneous between the different autograft centers. (cf Desai et al. "Anticoagulation practices in total pancreatectomy with autologous islet cell transplant patients: an international survey of clinical programs", *Transpl Int.* 2021, could be added to the bibliography)
 - line 196: (256 ± 93 IEQ vs. 214 ± 190 IEQ) : please correct the units in 103 IEQ
 - In this sentence you report no difference between groups in term of islet yield isolated. Maybe this yield should be reported in terms of IEQ per grams of pancreas isolated because pancreas weight in rescue totalization surgery and primary total pancreatic resection are different. Furthermore, the Milan team had shown that the number of isolated islets in case of a two-step rescue procedure was less important.
 - Line 232: you should add this major reference that illustrates your point of view "The Characterization and Prediction of ISGPF Grade C Fistulas Following Pancreatoduodenectomy" , McMillan, *J Gastrointest Surg*, 2016
- In this registry of patients with postoperative grade C pancreatic fistula, retrospective analysis of patients who should have benefitted from postoperative chemotherapy (55.7% of the patients in the cohort), in these patients adjuvant chemotherapy within the recommended time was delayed in 25.6% of cases and was never delivered in 67.4% of cases

Reviewer #2 (Remarks to the Author):

"Quality of life and metabolic outcome after total pancreatectomy and simultaneous islet autotransplantation – potential for a paradigm shift in pancreas surgery?" presents a series of islet autotransplants performed under a variety of indications either at the time of total or completion pancreatectomy. The authors report excellent metabolic outcomes for this procedure, and importantly, significantly less morbidity when total pancreatectomy was the primary surgical

operation. These results have implications for considering total pancreatectomy as the initial operation when there is a high likelihood of requiring a completion surgery to reduce morbidity, and as well support processing completion pancreatectomy specimens for islet isolation that can provide important benefit for post-pancreatectomy metabolic control and quality-of-life. Strengths include the excellent islet isolation results in terms of both quantity of islets and purity, including from the completion pancreatectomy specimens, that are reflected in the clinical measures of metabolic control, and the inclusion of quality-of-life assessment in comparison to a well-matched control group that underwent total pancreatectomy without islet autotransplantation. Weaknesses include the inherently small sample size for an understandably infrequent indication, but nonetheless support further consideration toward broadening the indication for islet autotransplantation beyond chronic pancreatitis. A few minor points for consideration include:

- Insulin independence is defined with the requirement of HbA1c <7.5%, yet post-prandial glucose <7.8 mmol/l, which are not commensurate. I would consider at least HbA1c <7.0% as has been included in the consensus definition of treatment success for beta-cell replacement therapy (<https://pubmed.ncbi.nlm.nih.gov/29528967/>) and applied in the setting of islet autotransplantation (<https://pubmed.ncbi.nlm.nih.gov/33020957/>).
- It is notable that all recipients of islet autotransplantation avoided hypoglycemia episodes that have been reported in other series. This might deserve a few words of discussion, especially if there may be something in particular about the gastrointestinal reconstruction that may have helped to avoid the predominantly alimentary hypoglycemia that has been described (reviewed in <https://pubmed.ncbi.nlm.nih.gov/30541144/>).

Reviewer #3 (Remarks to the Author):

The authors are to be commended on this work. IAT is a technology that clearly needs broader application for several reasons. First and foremost, as their work attests, the preservation of endogenous insulin secretion is important to glycemic control and to optimized quality of life. Moreover, for those patients in whom primary or secondary (completion) pancreatectomy is indicated, the avoidance of type 3c (pancreatogenic) diabetes is imperative. Publications of this type are critical to the continued expansion of IAT.

Several questions arise which would benefit from clarification:

- 1) in 17 of 52 considered/evaluated patients, IAT was not performed. This seems like a high percentage. The readership would benefit in understanding why such a large number of patients did not undergo IAT.
- 2) In 11 of 35 cases where IAT did proceed, the isolation was deemed 'not successful' and no IAT occurred. This seems like an extraordinarily high percentage. What was it about these patients that led to these outcomes. Is this issue of expertise of the isolation team? Were these isolation "failures" more likely to occur early in this series? Were they more likely associated with a particular surgical team, or variation of a surgical technique? Much more detail is needed to understand this data point.
- 3) Criteria for stopping infusion (page 6 line 176) seem odd, as did the absence of any rise in portal pressure in all of the other cases. In our experience with over 160 cases of IAT, portal pressures always rise. Moreover, the University of Minnesota has published its detailed criteria for stopping an

infusion, and these seem to be widely accepted. What was the tissue volume in all of these infusions?

Point-by-point response to the referees

1st Revision of Manuscript "Quality of life and metabolic outcome after total pancreatectomy and simultaneous islet autotransplantation - potential for a paradigm shift in pancreas surgery?"

We are grateful to the reviewers for the time they spent on our manuscript and their thoughtful and constructive comments / criticism. We have rewritten and updated the manuscript and changes are shown in red in the revised version. Below we summarize the reviewers' criticism and outline our responses with references to the modified text for more details.

Specific responses to reviewer comments:

Reviewer #1:

The authors report their experience in the emerging field of autologous islet transplantation after pancreatectomy.

This retrospective study compares 14 patients treated with this complex one-stage procedure with 10 patients who received this treatment in two stages, i.e. after completion of a pancreatectomy for complications of the first procedure.

Remarks :

- please specify in the method if the matching procedure was random or selected with or without a propensity score

The retrospective study describes a small and highly heterogenous cohort of patients where a matching procedure is not applicable. A matching procedure was only applied for assessment of Quality of Life. Here, an equal number of patients that underwent standard surgical treatment without islet autotransplantation resulting in total pancreatectomy and insulin deficiency were identified and matched with regard to age, gender, malignant/non-malignant underlying disease. A propensity score was not calculated and would not be applicable due to the small case number.

- line 68 with the Balzano et al reference you could add the Sutherland et al article "Total Pancreatectomy and Islet Autotransplantation for Chronic Pancreatitis" (JACS 2011)

The reference has been included (page 2).

**University Hospital
Carl Gustav Carus
at the Technische Universität
Dresden**
Institution under public law of the free
state of Saxony

Visitor Address:
Fetscherstraße 74
01307 Dresden
Telefon 0351 458 -0

Board:
Prof. Dr. med. D. M. Albrecht
(CEO, Speaker)
Katrin Erk (CFO)

Chairman of the Supervisory Board:
Univ.-Doz. Dr. G. Brunner

Bank Accounts:
Commerzbank
IBAN DE68 8508 0000 0509 0507 00
BIC DRES DE FF 850

Ostächsische Sparkasse Dresden
IBAN DE28 8505 0300 3120 1377 81
BIC OSDD DE 81

Deutsche Kreditbank AG
IBAN DE78 1203 0000 0011 2073 70
BIC BYLADEM1001

USt-IDNr.: DE 140 135 217
USt-Nr.: 203 145 03113

- We have seen that in the cohort out of 35 isolated pancreas only 24 could be transplanted. It would be very useful to specify the criteria used for the release of the islet batches, in particular in terms of purity / viability and number of islets. We understand in the "results" section that an insufficient number of 100 IEQ / kg was used as a limit but please specify these criteria in the methods.

We thank the reviewer for this important advice. The methods section has been complemented accordingly (page 4).

In fact, beyond sterility testing, no strict release criteria regarding islet yield, purity and viability have been defined for the autologous setting. This approach is in accordance with other center practice. While the predictive value of transplanted islet mass is well defined in allogeneic islet transplantation (e.g. Friberg et al., Transplantation. 2012 Mar 27;93(6):632-8), there is no sufficient data in islet autotransplantation to clearly define the cut-off. Balzano and Piemonti propose an orienting benchmark of 50 IEQ/kgBW for an insufficient islet mass (Balzano and Piemonti, Curr Diab Rep 2014, 14:512). Overall, to date it remains an individual decision balancing the potential benefit through even a minimal functional islet mass and the risk of the islet infusion procedure.

The apparently high number of 11 pancreata that resulted in a non-transplantable islet preparation, reflects our policy to rather liberally go for an isolation attempt. We learned that the subjective macroscopic impression of the resected pancreas tissue not always correlates with the isolation outcome. Therefore we currently rather risk a technical isolation failure than missing out on a potential transplantable islet preparation. Increasing multicenter experience will potentially allow for better definitions of predictive patient and pancreas characteristics.

- please specify in the methods the anticoagulation protocol used for the intraportal autograft. Indeed, we could read in a very remaining article that the modalities of anticoagulation were heterogeneous between the different autograft centers. (cf Desai et al. "Anticoagulation practices in total pancreatectomy with autologous islet cell transplant patients: an international survey of clinical programs", Transpl Int. 2021, could be added to the bibliography)

This aspect is in fact critical and we thank reviewer for raising this issue. The anticoagulation protocol (addition of Heparin to the islet preparation and continuous i.v. heparin starting immediately postoperatively) was included in the methods section of the revised manuscript. Also the reference of Desai et al. was included in the bibliography (page 4).

- line 196: (256 ± 93 IEQ vs. 214 ± 190 IEQ) : please correct the units in 103 IEQ

The numbers have been corrected in the revised version (page 8).

- In this sentence you report no difference between groups in term of islet yield isolated. Maybe this yield should be reported in terms of IEQ per grams of pancreas isolated because pancreas weight in rescue totalization surgery and primary total pancreatic resection are different. Furthermore, the Milan team had shown that the number of isolated islets in case of a two-step rescue procedure was less important.

We totally agree with this comment and the additional information on islet yield/g pancreas was added in the revised manuscript (page 8 and table 2). In fact, the tissue mass in salvage pancreatectomy patients is generally lower and the yield is relatively higher. This finding is only surprising in the first view, since the majority of islets is located in the pancreas tail which is available for isolation in the salvage pancreatectomy group. This finding has also been reported by others.

- Line 232: you should add this major reference that illustrates your point of view "The Characterization and Prediction of ISGPF Grade C Fistulas Following Pancreatoduodenectomy" , McMillan, J Gastrointest Surg, 2016
In this registry of patients with postoperative grade C pancreatic fistula, retrospective analysis of patients who should have benefitted from postoperative chemotherapy (55.7% of the patients in the cohort), in these patients adjuvant chemotherapy within the recommended time was delayed in 25.6% of cases and was never delivered in 67.4% of cases

We thank the reviewer for this comment. The reference has been added to the revised manuscript (page 2).

Reviewer #2 (Remarks to the Author):

"Quality of life and metabolic outcome after total pancreatectomy and simultaneous islet autotransplantation – potential for a paradigm shift in pancreas surgery?"
presents a series of islet autotransplants performed under a variety of indications either at the time of total or completion pancreatectomy. The authors report excellent metabolic outcomes for this procedure, and importantly, significantly less morbidity when total pancreatectomy was the primary surgical operation. These results have implications for considering total pancreatectomy as the initial operation when there is a high likelihood of requiring a completion surgery to reduce morbidity, and as well support processing completion pancreatectomy

specimens for islet isolation that can provide important benefit for post-pancreatectomy metabolic control and quality-of-life. Strengths include the excellent islet isolation results in terms of both quantity of islets and purity, including from the completion

pancreatectomy specimens, that are reflected in the clinical measures of metabolic control, and the inclusion of quality-of-life assessment in comparison to a well-matched control group that underwent total pancreatectomy without islet autotransplantation. Weaknesses include the inherently small sample size for an understandably infrequent indication, but nonetheless support further consideration toward broadening the indication for islet autotransplantation beyond chronic pancreatitis. A few minor points for consideration include:

- *Insulin independence is defined with the requirement of HbA1c <7.5%, yet post-prandial glucose <7.8 mmol/l, which are not commensurate. I would consider at least HbA1c <7.0% as has been included in the consensus definition of treatment success for beta-cell replacement therapy (<https://pubmed.ncbi.nlm.nih.gov/29528967/>) and applied in the setting of islet autotransplantation (<https://pubmed.ncbi.nlm.nih.gov/33020957/>).*

We thank the reviewer for pointing at that discrepancy. In fact, the definition of insulin independence was mistyped and has been corrected in the revised manuscript (page 5).

- *It is notable that all recipients of islet autotransplantation avoided hypoglycemia episodes that have been reported in other series. This might deserve a few words of discussion, especially if there may be something in particular about the gastrointestinal reconstruction that may have helped to avoid the predominantly alimentary hypoglycemia that has been described (reviewed in <https://pubmed.ncbi.nlm.nih.gov/30541144/>).*

This important aspect has been included in the discussion section of the revised manuscript (page 10).

Reviewer #3 (Remarks to the Author):

The authors are to be commended on this work. IAT is a technology that clearly needs broader application for several reasons. First and foremost, as their work attests, the preservation of endogenous insulin secretion is important to glycemic control and to optimized quality of life. Moreover, for those patients in whom primary or secondary (completion) pancreatectomy is indicated, the avoidance of

type 3c (pancreatogenic) diabetes is imperative. Publications of this type are critical to the continued expansion of IAT.

Several questions arise which would benefit from clarification:

1) in 17 of 52 considered/evaluated patients, IAT was not performed. This seems like a high percentage. The readership would benefit in understanding why such a large number of patients did not undergo IAT.

The initial number of 52 patients that were evaluated includes all patients that were scheduled for pancreas surgery. Those were screened to assess whether they were eligible for IAT. Main reasons for excluding patients was pre-existing insulin dependent diabetes, the presence of multifocal pancreatic neoplasm in preoperative imaging or intraoperative evaluation or insufficient or highly fibrotic/necrotic pancreas tissue unsuitable for islet isolation. In addition to the listing of inclusion/exclusion criteria (page 3), these points have been clarified in the results section of the revised manuscript (page 7).

2) In 11 of 35 cases where IAT did proceed, the isolation was deemed 'not successful' and no IAT occurred. this seems like an extraordinarily high percentage. What was it about these patients that led to these outcomes. Is this issue of expertise of the isolation team? Were these isolation "failures" more likely to occur early in this series? Were they more likely associated associated with a particular surgical team, or variation of a surgical technique? Much more detail is needed to understand this data point.

The apparently high number of 11 pancreata that resulted in a non-transplantable islet preparation, reflects our policy to rather liberally go for an isolation attempt. We learned that the subjective macroscopic impression of the resected pancreas tissue not always correlates with the isolation outcome. Therefore we currently rather risk a technical isolation failure than missing out on a potential transplantable islet preparation. However, the number of "failed" islet isolations that did not end up to be transplanted decreased significantly over the years and confirms an internal learning curve with regard to a better prediction of tissue suitability for islet isolation and an increasing multicenter experience will potentially allow for more clear definitions of predictive pancreas characteristics.

As hypothesized by the reviewer, the isolation was always performed by the same and highly experienced isolation team. Also the surgical team that performs these complex procedures allow for general consistency in surgical techniques. However, it will be worthwhile to document and retrospectively analyse potential

technical variations and surgical specifics more closely. We thank the reviewer for this recommendation.

In the revised version of the manuscript, those points were addressed in more detail (page 7).

3) Criteria for stopping infusion (page 6 line 176) seem odd, as did the absence of any rise in portal pressure in all of the other cases. In our experience with over 160 cases of IAT, portal pressures always rise. Moreover, the University of Minnesota has published its detailed criteria for stopping an infusion, and these seem to be widely accepted. What was the tissue volume in all of these infusions?

The criteria for stopping infusion is defined as “delta” from initial portal venous pressure of < 50% as also used by others (Balzano et al., AJT 2019). The rationale behind this rather strict approach compared to e.g. the Minnesota group where infusion is only stopped if the intraportal pressure exceeds 25 cm H₂O, is the often critical situation of patients particularly in the salvage pancreatectomy group. In this critically ill cohort we do not want to compromise about any additional stress. Due to the fact that the majority of preparations was purified prior to transplantation with a mean tissue volume of $2 \pm 2,6$ ml we were able to keep the portal venous pressure stable or within the allowed range (< 50% increase of basic value) in all cases besides one.

REVIEWERS' COMMENTS:

Reviewer #1 (Remarks to the Author):

Authors have answered satisfactorily
to all previous comments
I have no further remark

Reviewer #2 (Remarks to the Author):

Much improved. No further concerns or suggestions.

Reviewer #3 (Remarks to the Author):

Changes and clarifications made were reviewed in detail. Manuscript is acceptable for publication in its current form. no edits or revisions to suggest.